# Towards Extracellular Vesicles in the Treatment of Epidermolysis Bullosa

**DOI:** 10.3390/bioengineering12060574

**Published:** 2025-05-27

**Authors:** Aaron Gabriel W. Sandoval, Evangelos V. Badiavas

**Affiliations:** 1Harvard Medical School, Boston, MA 02115, USA; 2Aegle Therapeutics, Woburn, MA 01801, USA; ebadiavas@med.miami.edu; 3Dr. Phillip Frost Department of Dermatology and Cutaneous Surgery, University of Miami Miller School of Medicine, Miami, FL 33136, USA

**Keywords:** extracellular vesicles, exosomes, epidermolysis bullosa, wound healing, skin regeneration, mesenchymal stem cells

## Abstract

Epidermolysis bullosa (EB) is a debilitating genetic skin disorder characterized by extreme fragility, chronic wounds, and severe complications, particularly in its most severe form, recessive dystrophic EB (RDEB). Current treatments focus on symptomatic relief through wound care and pain management, with recent FDA approvals of Vyjuvek and Filsuvez providing new but limited therapeutic options. However, emerging research highlights the potential of extracellular vesicles (EVs) derived from mesenchymal stem cells as a promising approach to address both the symptoms and underlying pathology of EB. EVs function as carriers of bioactive molecules, modulating inflammation, promoting tissue regeneration, and even delivering functional type VII collagen to RDEB patient cells. Unlike whole-cell therapies, EVs are non-immunogenic, have greater stability, and avoid risks such as graft-versus-host disease or tumorigenic transformation. Additionally, EVs offer diverse administration routes, including topical application, local injection, and intravenous delivery, which could extend their therapeutic reach beyond skin lesions to systemic manifestations of EB. However, challenges remain, including standardization of EV production, scalability, and ensuring consistent therapeutic potency. Despite these hurdles, EV-based therapies represent a transformative step toward addressing the complex pathology of EB, with the potential to improve wound healing, reduce fibrosis, and enhance patient quality of life.

## 1. Introduction

Epidermolysis bullosa (EB) is a family of genetic skin conditions that are largely characterized by extremely fragile skin, which often leads to significant blisters, erosions, intractable itching, and excruciating pain. A unifying characteristic of these disorders is the fact that they each begin with mutations in the structural proteins that play a key role in maintenance of integrity in the skin and other epithelial tissues across the body. Although EB is most readily visible on the skin, any organ lined by epithelium can be affected, and extracutaneous manifestations of the disease can often include involvement of the eye, airway, and gastrointestinal or genitourinary tracts [1]. EB can be classified into four major subtypes based upon the specific causative mutations and level of separation within the layers of the skin: epidermolysis bullosa simplex, junctional epidermolysis bullosa, dystrophic epidermolysis bullosa (DEB), and Kindler syndrome. Among them, DEB, specifically the recessive variant (RDEB) which requires two defective copies of the causative gene to manifest, is generally considered the most severe EB subtype, given its profound impact on the integrity of the skin, overall quality of life, and mortality [2].

RDEB is classified as a rare disease with an estimated prevalence of 2 to 6 per million live births globally, and symptoms manifest at birth [3]. RDEB is a monogenic disorder caused by mutations in the COL7A1 gene that codes for type VII collagen (C7). C7 acts as an essential component of anchoring fibrils which serve to maintain the structure of the dermal–epidermal junction. In RDEB, C7 is either dysfunctional, reduced, or entirely absent, leading to separation of the epidermis from the dermis, which results in chronic wounds and subsequent scarring. RDEB leads to further morbidity in the form of severe pain, itch, infection, nutritional deficiencies (as a result of esophageal strictures), fusion of fingers and toes, and other ocular, gastrointestinal, and genitourinary involvement [4]. RDEB patients are highly predisposed to squamous cell carcinoma (SCC), with risk of developing SCC estimated at 76%, and mortality from SCC reaches 84% by age 40 [5]. As a result, the life expectancy for patients with RDEB is shortened, with many patients not surviving to the age of 40 [6]. Patients with RDEB often suffer from a more aggressive, deadly form of SCC compared to the form of SCC observed in the general population caused by UV sunlight exposure. SCC represents the leading cause of death in RDEB patients. The overall pathophysiology of SCC is not well understood and not completely explained by the genetic mutations inherent to RDEB; however, growing evidence suggests that the fibrotic, inflammatory milieu found in RDEB patients serves as a tumor microenvironment to encourage tumorigenesis [7]. RDEB patients report decreased quality of life due to impaired functioning and social activities, and beyond the debilitating nature of the disease, RDEB poses a significant socioeconomic and psychosocial burden on both the patients and their families. Daily wound care and dressing changes result in high healthcare costs along with caregiver burden and mental health challenges [5].

## 2. Current Non-Cellular Treatments for EB

The current standard of care for RDEB is primarily supportive, and focuses on symptomatic management through wound care with frequent dressing changes and cleansing to prevent infection and unroofing of blisters; pain and itch management using both nociceptive (NSAIDs, acetaminophen, opioids) and neuropathic (tricyclics, biofeedback, vibration, cooling) agents; and prevention of further complications through avoidance activities. However, none of these approaches can adequately address the underlying root cause of the disease and are, therefore, not considered to be curative [8]. Nevertheless, in recent years, the US Food and Drug Administration (FDA) has started to approve advanced therapies that aim to change this paradigm.

In May 2023, the FDA approved a gene therapy developed by Krystal Biotech called Vyjuvek (beremagene geperpavec-svdt) to treat wounds in patients suffering from DEB. This was a landmark decision, as Vyjuvek’s approval not only represented the first approved therapy to treat DEB but also the first ever topical gene therapy in any indication [9]. Vyjuvek uses a herpes simplex virus type 1 (HSV-1) vector which has been genetically modified to disable its ability to replicate and reduce its capacity to generate an immune response. The viral vector is designed to deliver normal copies of COL7A1 to open wounds. The HSV-1 genome does not integrate into the host cell DNA, instead remaining episomal and circumventing the concern for the insertional mutagenesis characteristic of many other viral gene therapy approaches. The viral genome is, however, diluted with each host cell division. As a result, the therapeutic effects of Vyjuvek are transient, necessitating repetitive dosing. Vyjuvek is compatible with a methylcellulose gel, allowing for topical administration in the outpatient setting and circumventing the need for general anesthesia, surgical operating rooms, and hospitalization [10]. In a phase 3 randomized controlled trial, 31 patients with DEB had size-matched wounds treated with either Vyjuvek or a placebo control every week for 26 weeks or until wounds had completely closed. Complete wound healing at 3 months or 6 months was statistically significantly improved in the Vyjuvek-treated wounds compared to placebo-treated wounds, and only mild side effects (pruritus, chills) were observed [11]. Nevertheless, Krystal’s study was impacted by various methodological limitations. The study had a small sample size, though this is an often unavoidable consequence of drug development for orphan diseases such as EB. Only a single patient in the study population had dominant DEB, limiting the generalizability of the findings to this patient population. The trial duration of 26 weeks was not sufficient to establish long-term safety or efficacy data. Furthermore, the intrapatient design of the study makes it difficult to assess systemic versus local effects of the therapy since each patient received both the placebo and treatment. Finally, this trial did not conduct skin biopsies to confirm restoration of C7 protein.

Though promising, this novel gene therapy is not without its drawbacks. Despite modifications made to the HSV-1 vector, inflammatory reactions are still possible, with a minority of patients reacting with erythematous rashes in response to Vyjuvek administration; however, none of the reported patients withdrew from the trial due to adverse events. However, based on the first author’s expanded clinical experience, some RDEB patients are not tolerant of the therapy, with significant inflammation and pain due to Vyjuvek precluding them from the therapy. Another concern is the possible development of an immune response to the HSV-1 vector or the novel C7 produced as a result of the therapy. Titers of antibodies against HSV-1 and C7 were increased in all patients who were seropositive prior to receiving Vyjuvek. Seroconversion occurred in 75% of patients without prior antibodies against HSV-1 and in 72% of patients without prior antibodies against C7 after 26 weeks of Vyjuvek treatment. Although immunogenicity did not seem to impact treatment efficacy during the limited study period, theoretically, antibodies against HSV-1 could lead to decreased efficacy over time, and antibodies against C7 could lead to the development of epidermolysis bullosa acquisita, an acquired autoimmune form of EB, often requiring immunosuppressive medications [12]. Indeed, consistent follow-up of patients will be required to ensure long-term safety, given Vyjuvek’s weekly administration schedule [10]. Also, the maximum weekly volume for patients over 3 years old is 1.6 mL, which only covers approximately 160 square cm [13], which is only a small fraction of the >30% total body area covered by wounds in approximately 60% of RDEB patients [14]. Finally, another concern with Vyjuvek is its substantial monetary cost, with a base cost of $300,000 per patient per year for the rest of the patient’s life. In the first year after its approval, estimated total expenditure for Vyjuvek in the US was $268 million, and given its weekly lifelong administration, Vyjuvek is projected to be the most expensive gene therapy currently marketed in the US [15].

Shortly after Vyjuvek’s approval, the FDA approved a second therapy for DEB in December 2023: the Chiesi Group’s Filsuvez (birch bark triterpenes) to treat junctional EB and DEB [16]. Filsuvez is a topical herbal gel composed of birch bark triterpenes, a historically traditional remedy derived from the bark of birch trees which has been shown to accelerate wound healing through upregulation of pro-inflammatory factors and increased keratinocyte migration [17]. In a phase 3 randomized controlled trial, 223 EB patients were enrolled, with 109 receiving Filsuvez and 114 receiving a placebo control gel. The primary endpoint of first complete target wound closure within 45 days was higher in the Filsuvez group compared to the control group. However, rates of severe adverse events were over twice as likely in the Filsuvez group, and one patient in the Filsuvez group did withdraw from the study as a result of a serious related adverse event (wound hemorrhage, probably related and wound infection, unlikely related) [18]. Like Krystal’s study, Chiesi’s study results should be interpreted in the context of its methodological limitations. Among patients with complete target wound closure within 45 days, 67% of patients in the treatment arm versus only 40% of patients in the control arm attended their appointment for assessment. Furthermore, only patients with RDEB demonstrated a statistically significant benefit from Filsuvez, whereas DDEB and JEB patients did not benefit from this therapy, which should theoretically be agnostic of EB subtype based on its putative mechanism. Like Krystal’s trial, Chiesi’s use of a 90-day assessment period constrains understanding long-term effects of the therapy. Finally, important secondary endpoints such as differences in procedural pain associated with dressing changes and skin activity, as measured by EBDASI [19], an index for grading EB severity, were not found to be statistically significant.

Moreover, Filsuvez’s pathway to FDA approval was not straightforward. In November 2021, the FDA extended the review of Filsuvez to analyze additional data provided by the company, and the FDA also submitted a new information request at that time. Then, in February 2022, the FDA denied Filsuvez’s approval, citing the necessity for additional confirmatory evidence to prove the therapy’s efficacy [20]. A tube of Filsuvez can only be used once after being opened; then the remainder must be discarded [21], which can become extremely expensive for patients with daily dressing changes. Furthermore, a key drawback to this therapy is that, although it accelerates wound closure, Filsuvez does not ameliorate the causative genetic defect that characterizes EB; thus, any wounds that do close do not contain functional protein, and wounds are theoretically just as likely to recur in the treated areas as a result. Thus, although Filsuvez may serve as a symptomatic treatment, it does not address the root cause of EB.

## 3. Cellular Therapies for EB

### 3.1. Ex Vivo Strategies to Treat EB

Despite the approval of two therapies for DEB, both therapies have significant drawbacks as described above, and novel therapeutic modalities will be required to fully treat EB. One avenue of active investigation is cell-based approaches, and a number of different cell types have been explored to treat EB [22]. Ex vivo gene therapy approaches entail correcting the causative mutation in allogeneic or autologous cells and subsequent transplantation of the cells back onto the patient. A landmark study demonstrated the feasibility of this concept through regeneration of the epidermis of a patient with JEB using autologous transgenic LAMB3 mutation-corrected keratinocyte cultures [23]. Results were durable 5 years post-treatment, likely due to genetic modification of an epidermal stem cell population that self-renews [24]. Similar trials were designed to treat RDEB, but a significant obstacle has been the large size of the COL7A1 gene, which has a negative impact on viral gene therapy vectors, limiting manufacture and transduction efficiency [25].

One phase 1 trial sponsored by Abeona Therapeutics corrected the COL7A1 gene in autologous keratinocytes with a retroviral vector and then grafted epidermal sheets onto the wounds of four patients; however, wound healing response was variable and declined over the course of a year [26]. In another phase 1/2 trial with adult participants, this therapy demonstrated long-term safety and efficacy, with improved wound healing 5 years post-transplant [27].

Despite these results, Abeona has faced regulatory hurdles. In September 2019, the FDA placed a clinical hold on the phase 3 trial due to unclear transport stability of the product [28], and in April 2024, the FDA rejected the treatment due to manufacturing issues [29], highlighting the complicated nature of developing an autologous ex vivo cell-based gene therapy comprising several manufacturing and transportation steps. Finally, in April 2025, the FDA approved this therapy, called Zevaskyn, based on phase 3 trial results demonstrating statistically significant wound closure and pain reduction at 6 months. Significantly, Zevaskyn represents the first single-dose therapy for RDEB, as the other two previously approved therapies, Vyjuvek and Filsuvez, must be reapplied regularly. Lasting improvement after just a single transplant with Zevaskyn was observed in patients from the phase 1/2 trial over a median follow-up period of almost 7 years [30]. Although formal publication of Abeona’s phase 3 trial results was not available at the time this review was written, analysis of the trial results submitted on clinicaltrials.gov reveals several methodological limitations [31]. This open-label trial lacked any placebos or blinding, which increases risk of bias, especially in subjective measures such as pain or healing appearance. As is common in EB trials, the small sample size of 11 patients limits generalizability, and the endpoint at 24 weeks limits long-term safety and durability analysis of the therapy. Finally, Abeona’s study utilized a primary endpoint of 50% healing from baseline, which is a much lower bar than the 100% complete wound closure endpoints set by Krystal and Chiesi for their own trials.

Another approach sponsored by Castle Creek Biosciences targets fibroblasts for ex vivo gene correction with a lentiviral vector. In a phase 1 study, autologous fibroblasts were genetically modified, then intradermally injected. Though safe, no new anchoring fibrils were detected, and repeat injections would likely be necessary for long-term efficacy [32]. Despite promising results, ex vivo gene therapy approaches which leverage harvesting, genetic modification, expansion, and transplantation of cells back to the patient can be quite costly, given all the steps involved in manufacturing. Indeed, Zevaskyn’s cost is currently $3.1 million, although Abeona has proposed an outcomes-based payment plan that reimburses payers if patients treated with Zevaskyn require retreatment within a 3-year period [33].

The aforementioned therapies fail to address systemic manifestations of EB beyond the skin; thus, other cell types have been explored for their therapeutic potential. Preclinical work in mouse models has shown that bone marrow-derived cells participate in skin wound healing. GFP-labeled bone marrow transplants were performed in non-GFP mice, and after wounding, GFP-positive cells were found in the epidermis, dermis, and accessory appendages (vessels, glands, hair follicles) of healed skin [34]. In a mouse model of RDEB, a transplant of wildtype bone marrow cells homed to wounded skin and improved skin integrity with production of C7 and anchoring fibrils [35]. Later work showed that a population of mesenchymal cells derived from mesenchymal stem/stromal cells (MSCs) in the bone marrow are mobilized to migrate and help regenerate skin in RDEB mice [36]. A systematic review of the literature found that a total of 55 patients with severe EB have been treated either by bone marrow transplant or bone marrow-derived MSC therapy, with 53 patients demonstrating improved wound healing, although 3 died of sepsis [37]. Despite its promise, there are several risks associated with bone marrow transplantation. The immunosuppressive regimens can be extremely toxic and lead to opportunistic infections, and graft-versus-host disease can occur, in which the donor cells attack the recipient’s body [38]. Thus, the potential benefits of bone marrow transplant must be carefully weighed against the risks.

### 3.2. MSC Treatments for EB

As MSCs are thought to play a central role in the therapeutic efficacy of bone marrow transplantation for EB, focus has shifted to isolating and administering these cells. MSCs are found in the bone marrow but also adipose or umbilical cord tissue. They are characterized by their trilineage differentiation potential into osteoblasts, chondrocytes, and adipocytes [39]. MSCs have potential clinical utility given their propensity to stimulate angiogenesis, dampen inflammation, and recruit tissue-specific progenitor cells to encourage tissue regeneration [40]. Given this ability to encourage wound healing, MSCs have been explored to treat EB. In a phase 1/2 clinical trial sponsored by RheaCell, 16 patients received three intravenous infusions of allogeneic dermal MSCs positive for the marker ABCB5. After 12 weeks, statistically significant decreases in clinical indices for EB activity and clinical outcomes were observed; however, two patients suffered from severe, serious hypersensitivity reactions which warranted termination of the therapy [41]. MSCs are generally considered to be immune evasive, meaning they can temporarily evade the host’s immune system through active modulation of the immune system through paracrine and cell–cell interactions [42]. Hypersensitivity reactions to intravenous infusion of HLA-unmatched, allogeneic MSCs, though rare, have been reported, and recent studies have shown that antibodies can be made against allogeneic MSCs, and immune rejection can still occur. Thus, it is possible that the adverse events in RheaCell’s trial resulted from immunological sensitization [42].

## 4. EVs to Treat EB

### 4.1. Advancements in MSC EVs to Treat EB

MSCs were first applied as a tissue replacement strategy; however, several studies have now shown that MSCs rarely engraft, given their short-lived viability post-administration [43]. Within 1 day post-infusion in mouse models, MSCs begin to activate apoptosis pathways and are engulfed by host macrophages [44]. MSCs are actually believed to cause therapeutic benefit through a ‘hit and run’ mechanism in which the MSCs still elicit lasting therapeutic benefit despite disappearing shortly after infusion. This finding led to the paracrine hypothesis that MSCs release certain pro-regenerative, anti-inflammatory signals. Further studies have shown that MSC-conditioned media alone confers similar benefits to MSCs themselves, encouraging tissue regeneration and modulating inflammation [45]. There is significant evidence to suggest that the paracrine mechanisms of MSCs could be driven both by soluble factors as well as extracellular vesicles (EVs), nanoparticles that are released from the endosomal compartment, which can be thought of as miniature versions of the cells which produce them.

The three classical categories of EVs, based on size, are exosomes, microvesicles, and apoptotic bodies, all of which contain lipids, proteins, and nucleic acids from the parent cell, which has led to further exploration of EVs as a potential therapeutic [46]. The smallest class of EVs is exosomes, which range from 30–150 nm. Exosome formation begins with endocytosis of the plasma membrane, forming early endosomes. Within early endosomes, intraluminal vesicles are formed via inward budding of the endosomal membrane, and as these intraluminal vesicles accumulate, multivesicular bodies result. When these multivesicular bodies fuse with the plasma membrane, the intraluminal vesicles are released into the extracellular space as exosomes. Microvesicles represent the second class of EVs, ranging in size from 0.1–1 µm. Microvesicles are formed through a process called ectocytosis, in which cytoskeletal rearrangement results in outward budding of the membrane. Finally, apoptotic bodies are the largest class of EVs, spanning 1–5 µm in size. They are formed during apoptosis, during which a dying cell breaks apart into apoptotic bodies which can contain nuclear fragments, cytoplasmic content, and even organelles such as mitochondria [47].

EVs can be purified through a variety of different methods, which can oftentimes bias the outcomes of downstream use cases; however, most EV purification protocols follow similar steps broadly (Figure 1). Generally speaking, conditioned medium from a cell type of interest is harvested from a cell culture disk or flask and transferred to centrifuge tubes. Next, the conditioned medium is differentially centrifuged in a stepwise manner using a variety of different speeds and durations to remove any remaining cells and other debris, resulting in a cleared conditioned medium. Finally, various methods can be used to enrich for the EVs, resulting in purified EVs. Isolation methods with high yield and low purity include polymer precipitation and membrane ultrafiltration. Methods with medium yield and purity include differential ultracentrifugation, field-flow fractionation, and size-exclusion chromatography. Finally, methods with low yield but high purity include gradient density ultracentrifugation, affinity isolation, flow cytometry, and microfluidics. In summary, several methods exist for EV purification, and the method chosen depends on the intended use case [48].

For therapeutic applications, size-exclusion chromatography (SEC) can be a favorable purification method. This method separates molecules based on hydrodynamic radius by passing a mixture through porous beads such that larger particles, such as EVs, do not enter the pores and thus elute earlier, whereas smaller molecules, such as proteins and nucleic acids, are trapped in the pores and elute later. SEC provides a balance of yield and purity, producing EVs with high expression of tetraspanins, which serve as surface markers of EVs. Furthermore, SEC can be easily scaled and adapted for use in most labs, is cost-effective, and results in excellent EV consistency and reproducibility, characteristics essential for therapeutic development [49].

EVs present several benefits over whole cell therapies. For instance, several preclinical and clinical studies have demonstrated that both autologous and allogeneic EVs are not immunotoxic [50]. Also, cell products can undergo malignant transformation during long-term in vitro culture [51], but this is not a concern for EVs, which are incapable of replication. Another issue associated with MSCs is the potential for emboli formation upon infusion, which is not likely to occur when utilizing EVs, given their nanosize [52]. Furthermore, EVs are much more stable than cell products, with the ability to be stored for several weeks at different temperatures or lyophilized without losing their bioactive properties. Hence, off-the-shelf applications to treat acute conditions are more feasible than with cell therapies, which must be thawed and often suffer from decreased cell viability [53]. Route of administration can vary widely, with EVs being administered topically, intradermally injected, or infused intravenously, or even inhaled to target different tissue types, which cannot be done for cell therapies that are unable to survive in certain environments [54].

Several key mechanisms working synergistically together underpin the extraordinary capabilities of EVs to maximize wound healing outcomes in injured states. Inflammation is the second stage of wound healing, immediately following hemostasis, and this process is essential to ensure proper removal of pathogens and other foreign material from the wound site. However, in certain diseased states, including EB, the inflammatory response can be dysregulated, resulting in prolonged, pathologic inflammation, which causes improper wound healing [55]. MSC EVs have been shown to modulate the immune response in a number of ways by promoting neutrophil phagocytosis and survival [56], as well as polarizing macrophages to the anti-inflammatory, pro-regenerative M2 phenotype, to encourage wound healing [57]. Moreover, MSC EVs, via impairment of dendritic cells, can reduce inflammatory effector T cells while increasing anti-inflammatory T regulatory cells [58]. Additionally, MSC EVs stimulate angiogenesis, cellular proliferation, and re-epithelization through pathways such as AKT and ERK signaling by delivering growth factors such as VEGF, EGF, and FGF [59]. Finally, hypoxia and ischemia caused by tissue injury generate oxidative stress, which can result in high levels of apoptosis and senescence at the site of injury. MSC EVs have been shown to carry antioxidant enzymes and miRNAs, which serve to protect cells from oxidative stress [60]. Moreover, apoptotic bodies specifically can transfer mitochondria, which are capable of neutralizing reactive oxygen species in damaged cells [61].

EVs represent a potentially promising therapy to treat EB, given the aforementioned advantages over cell therapy approaches and multifaceted mechanisms of encouraging wound healing. EVs are known to play a role in all stages of wound healing to encourage hemostasis, modulate inflammation, and incite proliferation and remodeling [62]. Similar to chronic, non-healing wounds, there is evidence to suggest that EB represents a systemic disease of persistent inflammation, and this inflammatory positive feedback loop must first be interrupted to allow for proper wound healing to take place [63], further supporting the use of MSC EVs, which have inherently immunomodulatory capabilities. Furthermore, preclinical studies have demonstrated that BM-MSC EVs transport C7 protein and mRNA in vitro, stimulating RDEB patient-derived fibroblasts to produce new C7, suggesting that EVs could not only accelerate wound healing and control inflammation in RDEB patients but also address the genetic root cause of the disease [64]. EVs have been shown to mitigate organ fibrosis in a variety of preclinical models through inactivation of myofibroblasts and fibrinolysis of extracellular matrix [65], and this is particularly relevant in EB, which leads to significant morbidity as a result of debilitating scarring and contractures [66]. Also, since EVs can be delivered in a variety of modalities, they could potentially address the extracutaneous manifestations of EB if delivered in a systemic manner. Overall, such a multifactorial approach to address the different aspects of EB offered by EVs is not seen in any currently approved therapy (Table 1).

### 4.2. Challenges and Future Directions

Despite their promise, EVs are not without their drawbacks. EV manufacturing and purification protocols lack standardization, and as a result, batch-to-batch variability and reproducibility are substantial concerns. Moreover, due to their small size and heterogeneous nature, it has been difficult to establish methods to quantify or characterize EVs [67], which could lead to inconsistent treatment outcomes in future clinical trials. Different strategies are being explored to increase the potency and scalability of EV production [68]. Dose discrepancies differ based on the disease model, so one proposal has been to focus on the potency of the EVs in in vitro assays to help determine dosing [69]. The issues of batch-to-batch variability and reproducibility could conceivably be addressed through the use of a single donor cell line, and studies have shown that EVs derived from immortalized MSCs are functionally equivalent to EVs derived from primary EV cell lines in certain regenerative contexts [70]. Although the exact mechanism of action of EVs is still relatively poorly understood and requires further research, the argument could be made that a comprehensive understanding is not necessarily required so long as the EVs are safe and efficacious [71]. Furthermore, EVs have a short half-life [68], so an EV therapy for EB would have to be readministered regularly. Finally, as there are no currently FDA-approved EV therapies, novel regulatory hurdles unique to this treatment modality will likely emerge as EV therapies begin to enter the clinic for EB and other indications.

There are also potential regulatory hurdles and translational challenges in EV therapies that may be unique to EB, in contrast to other conditions. Given the rarity of the disease, patient recruitment can prove difficult, with EB trials often enrolling fewer patients than trials for other skin conditions. Moreover, given the significant burden on quality of life, it can be viewed as unethical to administer the treatment to some patients and a placebo to others. Intrapatient controls, wherein different wounds on the same patient are treated with either the treatment or a placebo, have been used in the Vyjuvek and Zevaskyn trials, but this makes it difficult to discern systemic affects of the therapies. While clinical trials often restrict the use of other therapies beyond standard wound care measures to avoid confounding, as more EB therapies enter the market, it will become more difficult to recruit patients who are not concurrently using one of the approved EB therapies. Lastly, as EB often entails significant organ involvement beyond the skin, this disease can be more difficult to treat in clinical trials than other skin-limited conditions. For many of these reasons, the FDA created accelerated regulatory pathways such as the regenerative medicine advanced therapy (RMAT) designation [72] and incentives such as the rare pediatric disease Priority Review Voucher (PRV)—which has since been ended as of September 2024 [73]—to encourage therapy development for rare pediatric diseases like EB. Aegle Therapeutics is currently conducting a first-in-human Phase 1/2a clinical trial using EVs to treat EB, representing a key step towards realizing the potential of EVs to treat EB [74].

Although the majority of the discussion regarding EVs in this article has focused on the use of naïve EVs purified from conditioned media without significant modification, there have also been significant technological advancements in artificially engineering EVs to deliver certain cargo, creating or modifying nanoparticles with particles of interest, as engineered EV approaches provide several advantages in terms of manufacturing scalability when compared to the low yields associated with cell-secreted EVs [75]. EVs can carry diverse cargo, allowing for gene editing, gene silencing, and even transcriptional regulation, presenting certain advantages over traditional gene therapy approaches, given that EVs are immunologically neutral and have a low likelihood of inducing an immune response and subsequent seroconversion, which can be an issue with viral vector approaches [76]. Engineered EVs have been shown to serve as a functional collagen replacement therapy in preclinical models by delivering COL1A1 mRNA to induce collagen formation, and a similar approach could plausibly be used to deliver functional COL7A1 mRNA to RDEB patients [77]. Furthermore, EVs are being explored as a drug delivery platform, as they can be loaded with drugs of interest, augmenting EVs with molecules derived from other organisms or even synthetic molecules [78]. In the case of EB, for instance, MSC EVs could theoretically be loaded with molecules such as birch bark triterpenes, the active ingredient in Filsuvez, to further enhance wound healing outcomes. Currently, efforts to enhance engineered EVs are focused on maximizing EV production from donor cells, increasing the loading efficiency of cargo of interest into EVs, and improving tropism and cargo release at the target tissues [79].

## 5. Conclusions

EB has long been viewed as an untreatable disease managed primarily through supportive care; however, with the recent FDA approval of two therapies, with many more on the way, this paradigm is beginning to shift. Vyjuvek, the first redosable and topical gene therapy of any kind and the first EB therapy, as well as Filsuvez, the first therapy for multiple types of EB, have both provided EB patients with hope for an improved quality of life. Nevertheless, further work must be done to address the many facets of this devastating disease. A challenge that remains with EVs and many other EB therapies on the market and in development is that they largely address active wounds but are unable to prevent wounds from occurring. A true ‘cure’ for EB would likely require a gene therapy strategy targeting early-stage embryos to correct the genetic defect or some type of protein/gene replacement strategy that acts systemically to address the cutaneous and extracutaneous manifestations of EB. In the meantime, however, MSC-EVs may present a multifactorial approach to encourage wound healing, modulate inflammation, and deliver C7 mRNA and protein, to potentially alleviate the significant morbidity associated with EB and allow for improved quality of life for patients suffering from this tremendous disease. Nevertheless, further clinical trials will be necessary to verify these hypotheses and achieve the promise of EVs to treat EB.

## Figures and Tables

**Figure 1 bioengineering-12-00574-f001:**
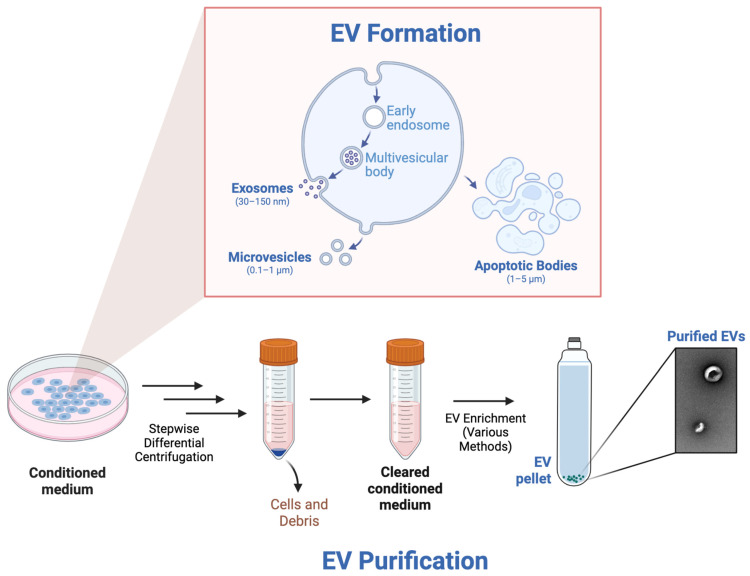
EV formation and purification.

**Table 1 bioengineering-12-00574-t001:** Comparison of EB therapeutic modalities.

Modality	Example	Immuno-modulatory	Functional Col7	Shelf-Stable	Acellular	Scalability	Ease of Admin.
In Vivo Gene Therapy	Modified HSV-1 vector	×	✓	×	✓	Medium	High (Topical)
Autologous Ex Vivo Gene Therapy	AAV-modified keratinocytes	×	✓	×	×	Low	Low (Transplant)
Allogeneic MSC Therapy	Dermal ABCB5+ MSCs	✓	✓	×	×	Low	Medium (IV infusion)
Small Molecule	Birch bark triterpenes	✓	×	✓	✓	High	High (Topical)
Allogeneic Extracellular Vesicles	Bone marrow-derived MSC EVs	✓	✓	✓	✓	Medium	High (Topical)

## Data Availability

No new data were created or analyzed in this study. Data sharing is not applicable to this article.

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
