# Peer review of "Towards Extracellular Vesicles in the Treatment of Epidermolysis Bullosa"

_bioengineering, 2025, doi:10.3390/bioengineering12060574_

Round 1
Reviewer 1 Report
Comments and Suggestions for Authors
The review is well written and focuses of various approaches to treat various skin diseases, in particular Epidermolysis bullosa.
Given the limited data so far with EV:s in the presented field, perhaps the conclusions about the future promise of EV are perhaps reaching a bit far pointing mainly to in vitro data (where the other modalities have clinical human data).
The conclusion could contain a bit more description of Vyjuvek, which after all is a successful drug and a first-in-class approval.
Author Response
Comment 1: The review is well written and focuses of various approaches to treat various skin diseases, in particular Epidermolysis bullosa.
Response 1: We appreciate the reviewer's positive assessment of our review.
Comment 2: Given the limited data so far with EV:s in the presented field, perhaps the conclusions about the future promise of EV are perhaps reaching a bit far pointing mainly to in vitro data (where the other modalities have clinical human data).
Response 2: Language surrounding the future promise of EVs was softened in the conclusion, and the necessity of further clinical trials to realize this promise was highlighted.
Comment 3: The conclusion could contain a bit more description of Vyjuvek, which after all is a successful drug and a first-in-class approval.
Both Vyjuvek and Filsuvez, the two currently approved EB therapies, were further highlighted in the conclusion.
Reviewer 2 Report
Comments and Suggestions for Authors
The authors reviewed the advancement and challenges of different treatments for epidermolysis bullosa (EB). The authors provided sufficient evidence to support the need for transitioning from cell-based therapies to extracellular vesicle-based therapies. The useof mesenchymal stem cells-derived EVs in phase 1/2 trial led by Aegle Therapeutics is exciting. However, there are several suggestions for enhancing discussion in the manuscript, especially for "EVs to Treat EB."
- There are many technological advancements in engineering EVs to deliver large cargoes. For instance, the research in PMID: 36635419 showed that EVs can deliver mRNA of COLA1 with their technology described in PMID: 31844155. Also, it is reported that EVs isolated from hTERT immortalized mesenchymal stem cells provided by ATCC showed similar function compared to primary MSC. The authors mentioned challenges of EV-based drug delivery such as "batch-to-batch" challenge in the end of section of "EVs to treat EB" but it is highly recommended to conclude the section with potential solutions to the challenges discussed.
- It is recommended to provide a schematic figure to illustrate the current treatments and potential treatments for EBs.
- For improving readability, it is recommended to break section 3 into subsections, such as "Ex Vivo strategy to treat EB" and "MSC treatment for EB". Similarly, section 4 could be divided into subsections like "Advancement in MSC EV to treat EB" and "Challenges and future direction to treat EB".
Author Response
Comment 1: There are many technological advancements in engineering EVs to deliver large cargoes. For instance, the research in PMID: 36635419 showed that EVs can deliver mRNA of COLA1 with their technology described in PMID: 31844155. Also, it is reported that EVs isolated from hTERT immortalized mesenchymal stem cells provided by ATCC showed similar function compared to primary MSC. The authors mentioned challenges of EV-based drug delivery such as "batch-to-batch" challenge in the end of section of "EVs to treat EB" but it is highly recommended to conclude the section with potential solutions to the challenges discussed.
Response 1: We appreciate this reviewer's suggestion to provide potential solutions to the drawbacks of EVs, and have incorporated these points as well as the cited references into the article.
Comment 2: It is recommended to provide a schematic figure to illustrate the current treatments and potential treatments for EBs.
Response 2: Other review articles in the literature have created comprehensive schematic figures which illustrate current and potential treatments for EB (for example, https://doi.org/10.2147/TCRM.S386923), and the focus of our article was to present the value of EVs in the context of these other modalities rather than to comprehensively review all EB treatments in development.
Comment 3: For improving readability, it is recommended to break section 3 into subsections, such as "Ex Vivo strategy to treat EB" and "MSC treatment for EB". Similarly, section 4 could be divided into subsections like "Advancement in MSC EV to treat EB" and "Challenges and future direction to treat EB".
Response 3: These suggestions were appreciated and incorporated into the revised manuscript.
Reviewer 3 Report
Comments and Suggestions for Authors
This manuscript explores the potential use of EVs in treating EB, a group of rare genetic skin disorders. It aims to summarize the current understanding of EB, limitations of existing therapies, and emerging strategies involving MSC-derived EVs.
While the topic is timely and highly relevant to the fields of regenerative medicine and translational research, the manuscript in its current form lacks the scientific rigor, critical analysis, and structural clarity expected for publication in a journal such as Bioengineering. The content remains largely descriptive and does not provide sufficient mechanistic insights, evaluative commentary, or original perspective. Major revisions are necessary to elevate this manuscript to a publishable standard.
Here are my detailed comments and suggestions:
- The manuscript summarizes existing findings without providing critical appraisal of the literature. There is no discussion of the limitations of cited studies, methodological challenges, or contradictory results.
- There is a need to identify gaps in the literature and suggest future directions in a more thoughtful, hypothesis-driven manner.
- The manuscript lacks in-depth discussion on key topics such as EV biogenesis, cargo loading, engineering strategies, and delivery mechanisms, which are crucial in a bioengineering journal.
- While therapeutic potentials of EVs are mentioned, there is no in-depth explanation of how EVs mediate healing or influence disease pathology in EB. Mechanistic pathways (e.g., immunomodulation, ECM remodeling, signal transduction) should be discussed with references to relevant studies.
- The structure of the manuscript is uneven. Some sections are overly detailed (e.g., EB subtypes), while others are too brief (e.g., clinical trials, challenges in EV-based therapy).
- The manuscript would greatly benefit from the inclusion of schematics or figures. Suggested additions include:
- A diagram of EV biogenesis and delivery pathways
- A table comparing traditional EB therapies with EV-based strategies
- A summary of current or completed clinical trials involving EVs
- There is only minimal mention of clinical trials involving EVs for EB or related skin conditions. A more comprehensive summary of ongoing trials, regulatory hurdles, and translational challenges is needed.
- The potential of EVs as drug delivery vehicles or gene-editing platforms is not discussed, which is highly relevant.
- Several key papers and recent reviews on EVs in dermatological or regenerative medicine contexts are missing.
- The manuscript is heavily reliant on secondary sources. Inclusion of more primary research articles would improve scholarly value.
Author Response
Comment 1: The manuscript summarizes existing findings without providing critical appraisal of the literature. There is no discussion of the limitations of cited studies, methodological challenges, or contradictory results.
Response 1: Rather than critically appraising the literature, the aims of this review were to summarize and present the work that has led to the development of EV therapies to treat EB and to explore the pros/cons of EVs to treat EB.
Comment 2: There is a need to identify gaps in the literature and suggest future directions in a more thoughtful, hypothesis-driven manner.
Response 2: The main gap in the literature is the lack of clinical trial results applying EVs to EB patients to determine whether the extensive preclinical findings in the literature will ultimately translate to the clinic. Our group is currently conducting a clinical trial to answer this question. To address the reviewers concerns, future directions were expounded upon through revisions made in the section titled 'Challenges and Future Directions'.
Comment 3: The manuscript lacks in-depth discussion on key topics such as EV biogenesis, cargo loading, engineering strategies, and delivery mechanisms, which are crucial in a bioengineering journal.
Response 3: There are already several publications in the literature which cover these topics in great depth, including publications from our own group (DOI: 10.1016/j.jid.2017.04.021), and we feel that such an in-depth discussion would go beyond the scope of our review, which was aimed at exploring the progression of research towards the use of EVs to treat EB. Discussion of manufactured/modified EVs versus naive cell-derived was added to the section titled 'Challenges and Potential Solutions'.
Comment 4: While therapeutic potentials of EVs are mentioned, there is no in-depth explanation of how EVs mediate healing or influence disease pathology in EB. Mechanistic pathways (e.g., immunomodulation, ECM remodeling, signal transduction) should be discussed with references to relevant studies.
Response 4: Several articles exist in the literature already covering the mechanistic pathways of how stem cells and their EVs mediate healing in EB (for example, this article published in Bioengineering: DOI: 10.3390/bioengineering10040422), and that is not the focus of our review article as it is detailed elsewhere in the literature in great detail.
Comment 5: The structure of the manuscript is uneven. Some sections are overly detailed (e.g., EB subtypes), while others are too brief (e.g., clinical trials, challenges in EV-based therapy).
Response 5: The section on challenges in EV-based therapy was expanded upon. The section on clinical trials is brief because there is only a single clinical trial currently being conducted using EVs to treat EB. This trial is being conducted by our group, and mention of our currently ongoing trial was added at the end of the article.
Comment 6: The manuscript would greatly benefit from the inclusion of schematics or figures. Suggested additions include:
- A diagram of EV biogenesis and delivery pathways
- A table comparing traditional EB therapies with EV-based strategies
- A summary of current or completed clinical trials involving EVs
Response 6: Other publications from our own group have already included similar figures.
- A diagram of EV biogenesis and delivery pathways - DOI: 10.1016/j.jid.2017.04.021
- A summary of current or completed clinical trials involving EVs - DOI: 10.1007/s00403-024-03232-5
- The focus of our review was not to explore the use EVs to treat any clinical indication but rather specifically for the treatment of EB.
We agree with this reviewer's comment that a table comparing the different EB treatments would be valuable to help readers better visualize the pros/cons of each approach and how they compare against one another, head to head, so we have added such a table to our article.
- A table comparing traditional EB therapies with EV-based strategies - A table was added to the revised manuscript comparing the different EB therapeutic modalities.
Comment 7: There is only minimal mention of clinical trials involving EVs for EB or related skin conditions. A more comprehensive summary of ongoing trials, regulatory hurdles, and translational challenges is needed.
Response 7: To our knowledge, there are not any completed clinical trials using EVs to treat EB, and we are currently conducting the only clinical trial using EVs to treat EB. The results from our group's trial are forthcoming, and the points raised by the reviewer will be addressed in that future publication. Nevertheless, mention of our ongoing trial, regulatory hurdles, and translational challenges were added to the section titled 'Challenges and Potential Solutions'.
Comment 8: The potential of EVs as drug delivery vehicles or gene-editing platforms is not discussed, which is highly relevant.
Response 8: This review was not meant as a comprehensive overview of EVs or gene editing therapeutic modalities, several of which already exist in the literature, but rather to highlight EVs as they relate to treating EB specifically.
Comment 9: Several key papers and recent reviews on EVs in dermatological or regenerative medicine contexts are missing.
Response 9: We feel that EVs in other contexts beyond EB goes beyond the scope of our review, and other articles in the literature cover these topics comprehensively, including previous publications from our own group: MSC EVs to treat chronic wounds (DOI: 10.1101/cshperspect.a041227) and EVs in dermatology (DOI: 10.1016/j.jid.2017.04.021).
Comment 10: The manuscript is heavily reliant on secondary sources. Inclusion of more primary research articles would improve scholarly value.
Response 10: Where possible and appropriate, we tried our best to include primary research articles.
Round 2
Reviewer 3 Report
Comments and Suggestions for Authors
I appreciate the authors' efforts to address the comments provided. The authors have offered defensible but sometimes minimalist responses. Several concerns, including critical analysis, mechanistic discussion, and broader context, were not fully addressed. While the scope justification is reasonable, it may limit the review’s appeal to a broader audience. I believe that further revisions are necessary to strengthen the manuscript’s scholarly impact.
Specific points for further consideration:
- The authors state that the goal was to summarize existing findings. However, even summary-style reviews benefit from brief mention of methodological limitations, contradictory findings, or study weaknesses. A summary without any critical lens diminishes the scholarly value.
- While the authors argue that detailed discussion is outside the review’s scope, even a concise paragraph overview or a simplified schematic would strengthen the manuscript, particularly for a bioengineering audience.
- The authors state that mechanistic pathways are discussed elsewhere. However, providing a brief summary of key mechanisms (a few sentences supported by references) would improve the paper’s depth and help readers better understand how EVs mediate therapeutic effects.
- The addition of the comparative therapy table is appreciated. However, even if similar figures exist elsewhere, including a simplified, targeted schematic would significantly enhance clarity and reader engagement.
- The authors declined to discuss EVs as drug delivery vehicles or gene-editing platforms, stating it is outside the scope. However, a short paragraph acknowledging these applications, even without extensive detail, would provide broader context and strengthen the translational relevance of the review.
Author Response
Comment 1: The authors state that the goal was to summarize existing findings. However, even summary-style reviews benefit from brief mention of methodological limitations, contradictory findings, or study weaknesses. A summary without any critical lens diminishes the scholarly value.
Response 1: Methodological limitations and study weaknesses of clinical trials were highlighted in the manuscript.
Comment 2: While the authors argue that detailed discussion is outside the review’s scope, even a concise paragraph overview or a simplified schematic would strengthen the manuscript, particularly for a bioengineering audience.
Response 2: A paragraph and a figure detailing both EV formation and purification were added to the manuscript.
Comment 3: The authors state that mechanistic pathways are discussed elsewhere. However, providing a brief summary of key mechanisms (a few sentences supported by references) would improve the paper’s depth and help readers better understand how EVs mediate therapeutic effects.
Response 3: A paragraph detailing key mechanisms of EV therapeutic effects was added to the manuscript.
Comment 4: The addition of the comparative therapy table is appreciated. However, even if similar figures exist elsewhere, including a simplified, targeted schematic would significantly enhance clarity and reader engagement.
Response 4: A figure detailing EV formation and purification was added to the manuscript.
Comment 5: The authors declined to discuss EVs as drug delivery vehicles or gene-editing platforms, stating it is outside the scope. However, a short paragraph acknowledging these applications, even without extensive detail, would provide broader context and strengthen the translational relevance of the review.
Response 5: A paragraph discussing EVs as drug delivery vehicles and gene-editing platforms was added to the manuscript.